# The Implication of Steel-Intensity-of-Use on Economic Development

**Harimukti Wandebori [1,*] and Murtyastanto [2]**

1. Sekolah Bisnis dan Manajemen, Institut Teknologi Bandung, Jl. Ganesha No. 10, Bandung 40132, Indonesia
2. Krakatau Steel, Jl. Industri No. 5, Cilegon 42435, Indonesia; murtyastanto.kp@krakatauposco.co.id
* Correspondence: harimukti@sbm-itb.ac.id

**Abstract:** This paper identifies statistical relationships between the Steel-Intensity-of-Use and the Gross Domestic Product per Capita based on the data of ten selected countries based on judgmental sampling from the World Steel Association. Economic data such as Gross Domestic Product, Gross National Income, Gross Domestic Product per Capita, government spending, investment, Export of Goods and Services, and manufacturing are obtained. Based on regression analysis and exploration of economic data, the relationship differs in terms of economic development stratification. We find a pattern of a waveform in terms of Correlation and Significance between Steel-Intensity-of-Use and the Gross Domestic Product per Capita along the country's economic classification. Each classification implies that the country needs to enhance the economy related to the required steel or non-steel-related industry and the accompanying government policy.

**Keywords:** steel-intensity-of-use; steel industry; economic classification; gross domestic product; economic and policy development

## 1. Introduction

Steel plays an essential role in the development of the manufacturing sector, for it is the material of choice for many elements of manufacturing. The authors of [1] mention that dynamic transitions in economics, with a key asset in the manufacturing context and political situation, is the driver that has the greatest impact on the achievement of sustainability. It is one of the drivers aimed at mitigating climate change in the context of the manufacturing industry [1].

The authors of [2] explain that growing the manufacturing sector may encourage other sectors to flourish. Besides that, a more significant contribution of the manufacturing sector to employment would boost the gross private savings and technology accumulation. Accordingly, ref. [2] mention that the manufacturing sector establishes an incentive for gross private saving and capital accumulation, the valuable determinants of economic growth through industrialization. Linkage and spillover effects are stronger in manufacturing than in agriculture or mining [3]. Linkage effects refer to the direct backward and forward linkages between different sectors, which create positive externalities for investments in given sectors. The externalities are associated with the entrepreneurial market entry [4], generating information about the country's latent comparative advantage. Spillover effects refer to the disembodied knowledge flows between sectors [3]. These are special cases of externalities that refer to investment in knowledge and technology. Linkage and spillover effects are presumed to be stronger within manufacturing than within other sectors except with services and agriculture [3].

In 2017, the world steel industry conditions improved, indicating a return to global steel demand and production [5], but steel market fundamentals weakened considerably since 2019. Steel production growth has turned negative in all regions except in Asia and the Middle East. Weakening global economic activity, uncertain prospects for steel demand growth, and the upturn in new capacity investments in some regions continue to cloud the outlook for the global steel market and excess capacity. The elucidation of this upheaval

phenomenon can be designated by the Intensity-of-Use hypothesis, which establishes the Steel-Intensity-of-Use (I-U) (i.e., per capita total steel consumption per Gross Domestic Product (GDP) per capita) [6]. It was initially introduced by the International Iron and Steel Association (now World Steel Association) but popularized in [7–9].

According to the I-U hypothesis, the intensity of steel use relies on economic development, but it is not believed to be linear [6]. It is because of the economic development cycle. There is a high industrialization phase growth in relatively poorer countries, implying a high quantity of I-U. As industrialization is saturated, it usually switches to services for developed countries, which means maturity to decline I-U as the GDP increases.

Steel is one of the prominent metals as it is widely used to develop industries, so there is a correlation between steel consumption and the economy [10]. Steel Intensity-of-Use follows the hypothesis. It has an inverted U-shaped along with GDP. There is a point of GDP per capita as the peak where the intensity of use starts to decline [11]. This condition is due to the fact that I-U for a country is a function of its product composition of output, which in turn depends on the GDP per Capita [12], while components of GDP comprise personal consumption, investment, government spending, and net exports [13].

The I-U hypothesis has been tested on several metals, for example, steel [9], aluminum, copper, lead, steel, zinc, and nickel [11,12,14–19]. Despite the existence of literature on I-U, particular research on the relationship between I-U and GDP per capita in terms of the stratification of economic development (from a low-income country to a high-income country) is still non-existent. The gaps are particularly related to the steel industry's roles in economic development and influences on other industries, industry differences among countries within similar phases of development, and required steel industry and related industry development projection based on correlations and significance.

This research reveals the relationship between the I-U and the GDP, differentiated into economic development stratification, which may differ from the previous studies. It also describes the economy, general industry, and steel industry and unravels I-U's required roles to develop the industries and transform the economy within economic development, which is not undertaken by the previous studies.

The main purpose of this paper is to elaborate on the I-U and its implication on economic development. How significantly does I-U in the related steel industries influence economic development? Are there any discrepancies among nations with different economic conditions regarding I-U's influences on economic development? What are industries, in relationships with the I-U and the GDP per Capita, to be developed in a country with particular economic development?

The revealed correlation and its significant level is distinguished based on the respective country's economic development. Its pattern is depicted as a waveform in terms of correlation and significance between Steel-Intensity-of-Use and the Gross Domestic Product per Capita along the country's economic classification. The position of a country in the waveform implies the roles of the steel industry in economic development, how it is transformed in related industries, and how it switches to the service industry along with economic development.

## 2. Literature Review

### 2.1. Steel Consumption and Economy

In 2019, apparent steel use comprised China (51.3%), Other Asia (9.9%), EU (9.0%), NAFTA (7.6%), India (5.7%), Japan (3.6%), CIS (3.3%), Other Europe (1.9%), and Others (7.6%) [20]. The economic activities which absorb high steel consumption in the forms of capital goods and infrastructures are the manufacturing and construction sectors [21]. In developed countries, economic activity indicated as Gross Domestic Product (GDP) does not experience high development or remain constant from year to year. In contrast, developing countries tend to proliferate [22].

In the beginning, developing countries started to build empirically where growth in steel consumption with high absorption occurred in the manufacturing and construction

industry sectors [23]. Meanwhile, at the mature stage or after becoming a developed country, steel consumption growth has stagnated or slowed down in steel consumption [24], shifting the development in other sectors such as the service industry and trade [23].

The author of [24] explains this phenomenon of consumption patterns and economic activity using I-U, defined as the demand or consumption of steel in a certain period per unit of GDP. The authors of [25] suggest an empirical and typical relationship in the form of an inverted "U" curve between the I-U and GDP per capita. The inverted "U" curve characteristic in Figure 1 indicates that the increase in steel use intensity is not proportional to GDP per capita. The increase in the I-U peaked quickly only after reaching a certain point of GDP per capita.

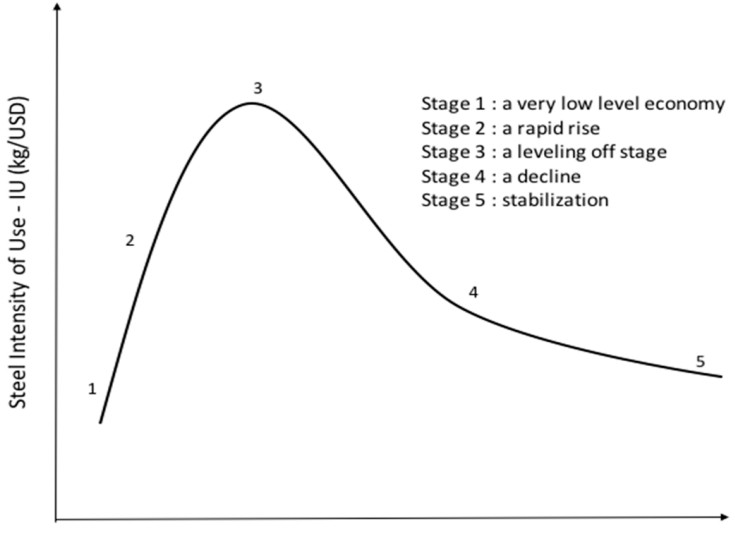

**Figure 1.** Steel Intensity Curve.

In contrast, the I-U tends to stagnate or even decline in the long term. The initial phase of the high intensity of using steel is called the industrialization phase. In the next phase, industrialization has shifted to sectors that intensively use services and technology. At the end of the phase, there will be a slowdown in the intensity of the use of steel, although it is still high in nominal terms. Japan is an example of a country that has experienced a decline in the intensity of steel use in the long run. It is due to a shift in the industry from manufacturing to industry based on services and technology.

As Figure 1 suggests, the state of I-U is different along with the GDP per Capita. Therefore, it is necessary to elaborate on the steel industry's role by exploring the interaction among I-U, related industries, and economic development.

### 2.2. I-U

I-U is defined as the ratio of per capita steel consumption to GDP per capita. If y denotes the consumption of steel per capita and x is the GDP per capita [26], then:

$$Steel\ Intensity = y/x,\ \text{in kg/unit GDP} \tag{1}$$

We use the basic traditional model and an extended version by [12] to estimate the I-U. The basic model assumes that the I-U for a country is a function of the country's product composition of output, which, in turn, depends on the GDP per Capita. This is functionally expressed as:

$$I - U_t = f\ (\text{GDP per Capita}_t) \tag{2}$$

The I-U is determined by the endogenous components of real GDP per Capita.

Based on [13], it is necessary to find the correlations between I-U and components of the GDP as personal consumption, investment, government spending, and net exports.

Considering the influences of components of GDP, we establish a new formula of I-U that accommodates them:

$$I - U_t = \mu_0 + \mu_1 \text{ Final Consumption per Capita}_t + \mu_2 \text{ Investment per Capita}_t + \mu_3 \text{ Export Goods and Services per Capita}_t - \mu_4 \text{ Import of Goods and Services per Capita}_t \tag{3}$$

$\mu_0$, $\mu_1$, $\mu_2$, $\mu_3$ and $\mu_4$ are parameters to be estimated. Final consumption per Capita comprises personal consumption and government spending per Capita.

### 2.3. Model Framework for Analysis of Steel Consumption Intensity

The paper is divided into two categories of frameworks to analyze the I-U as follows.

Firstly, it presents the correlations between I-U and real GDP per Capita and reveals the pattern of a waveform in terms of correlation and significance between Steel I-U and the GDP per Capita along the country's economic classification. Secondly, it uses the comprehensive model I-U which considers the relationship between I-U and GDP per Capita, Government Spending per Capita, Investment per Capita, Export of Goods and Services per Capita, and Manufacturing per Capita, along with the country's economic classification.

The analysis of the two categories is reinforcing and complementary to achieve the main purpose of this research.

### 2.4. Country Classification and Economic Stages of Development

The country classification is determined by the Gross National Income (GNI) per capita, which depends on its economic growth within its development stage [27]. GNI equals GDP plus net factor income from abroad. The author of [27] presents the concept of economic growth and argues that modernization's sequential economic steps can be identified within a society. He identifies five growth stages (traditional society, preconditions for take-off, take-off, drive to maturity, and age of high-mass consumption), which are linear and towards evolutional higher development in Figure 2. A country with a higher growth stage has more savings and investment in the industry [27].

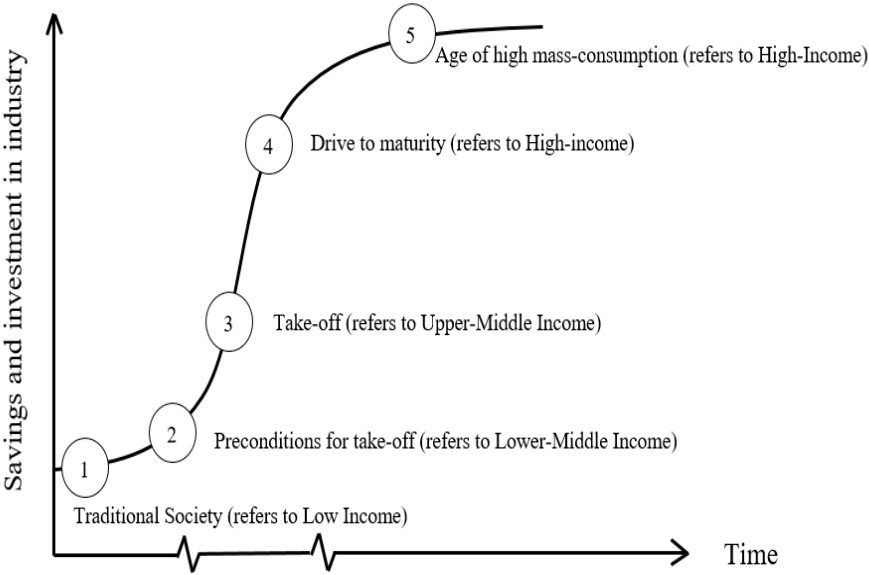

**Figure 2.** Rostow's five-stage model of development and World Bank Classification.

The five growth stages of Rostow's model are consistent with the World Bank's classification of the country's economic development, which comprises low-income, lower-middle-income, upper-middle-income, and high-income [28].

### 2.4.1. Traditional Society

Based on World Bank's classification [28], a low-income country has a Gross National Income (GNI) per capita of less than USD 1036. Low incomes are often associated with other characteristics, which are severe inequality, inadequate health care and education, high unemployment, heavy reliance on agriculture, and rapid population growth.

From this characteristic, the low-income countries in Rostow's economic development stages refer to the first stage of traditional society [27].

According to Rostow's model [27], in the stage of traditional society, the economic system is stationary and dominated by agriculture with traditional cultivating forms [29]. Productivity by man-hour work is lower compared to the following growth stages. Society characterizes a hierarchy, so there is a low vertical as well as social mobility. Manufacturing and service sectors are non-existent [30].

### 2.4.2. Precondition for Take-Off

Lower-middle-income economies are those with a GNI per capita, calculated using the World Bank Atlas method, of more than USD 1036 but less than USD 4045. It is the first stage of transformation of low-income countries that relies heavily on agriculture towards industry, requiring more steel consumption. Based on Rostow's model [27], it is a precondition for take-off from the transitional stage with a period of learning that will propel a country from a traditional society to a modern one [30]. During this stage, the rates of investment are thriving, and they initiate a dynamic development. This kind of economic growth is a result of the industrial revolution. As a consequence of this transformation, including agriculture development, the primary sector's workforces become redundant. A prerequisite for "the preconditions for take-off" is the industrial revolution, which may last for a century.

### 2.4.3. Take-Off

Upper-middle-income economies are those with a GNI per capita, calculated using the World Bank Atlas method, of more than USD 4046 but less than USD 12,535. In this classification, the countries develop and enhance the industries and consume more steel than the related industries, leading to national competitive advantage. Based on Rostow's model [27], it is the take-off stage of economic development. The beginning of take-off can usually be traced to a sharp stimulus, such as a political revolution, a technological innovation, or a particularly favorable or unfavorable shift in the international environment [30]. This stage is characterized by dynamic economic growth. The main characteristic of this economic growth is self-sustained growth which requires no exogenous inputs. A few leading industries can support development. Generally, "take-off" may last for two to three decades.

### 2.4.4. Drive to Maturity and Age of High Mass Consumption

High-income economies are those with a GNI per capita, calculated using the World Bank Atlas method, of USD 12,375 or more.

According to Rostow's model, high-income countries are in the stages of driving to maturity or the age of high mass consumption [27]. It is characterized by continual investments of 40 to 60 percent. Economic and technological progress dominate this stage. New forms of industries, such as neo-technical industries, emerge. Rostow defines this fourth stage as the period when a society has effectively applied the range of modern technology to its resources.

The age of high mass consumption is the final stage of Rostow's five-stage development model [19]. Very few countries have reached this stage [30]. It is a society of affluence and consumer power. Here, most parts of society live in prosperity, and people are offered both abundance and an assortment of choices.

## 3. Materials and Methods

We use published steel consumption and production data from the World Steel Association (WSA) from 2000 to 2021 [20]. Economic data such as GDP, Gross National Income (GNI), GDP per capita, government spending, investment, Export of Goods and Services, and manufacturing are from the World Bank. The classification of a country's economic development is based on the World Bank Classification and adapts Rostow's Five-Stage of Economic Development [27]. We put a real GDP per Capita and all components based on fixed price in 2015 from [31].

The case study is within the scope of global level. Using judgmental sampling, we selected China, India, United States, and Indonesia because they are the topmost populated countries. United States, Germany, Japan, South Korea, and China, except Russia, are the developed countries that experienced a decrease or erosion of the intensity of steel use in the last two decades. In contrast, Indonesia, India, Vietnam, and Malaysia are the developing countries that experienced a growing intensity of steel use.

The correlation between I-U and GDP per capita is classified as significant at the 0.05 level and significant at the 0.01 level. All data of the I-U and GDP per Capita, including the components, were processed as shown in Appendix A.

## 4. Results

### 4.1. I-U in Developing and Developed Countries

Table 1 shows that the intensity of steel use in developed countries, such as the United States, Germany, Japan, and South Korea, has experienced a diminishing intensity of steel use in the last two decades. It is low compared to world steel use intensity changes, which has increased to reach 16 percent in the last decade. Meanwhile, developing countries, such as Russia, Indonesia, Vietnam, and India, except Malaysia and China, experienced a growing intensity of steel use. The increasing intensity of steel use is supported by the success of industrialization and infrastructure development.

**Table 1.** I-U from 2000 to 2021 *.

| Country | Steel Intensity (Kg/USD) | | | |
| --- | --- | --- | --- | --- |
| | Average (2000–2021) | Average (2000–2010) | Average (2011–2021) | Change (%) |
| Indonesia | 0.014 | 0.013 | 0.015 | 19.1% |
| China | 0.066 | 0.067 | 0.065 | −2.3% |
| Germany | 0.012 | 0.013 | 0.011 | −11.7% |
| India | 0.035 | 0.048 | 0.069 | 45.0% |
| Japan | 0.016 | 0.018 | 0.014 | −19.4% |
| Malaysia | 0.034 | 0.039 | 0.030 | −23.3% |
| South Korea | 0.041 | 0.046 | 0.037 | −20.5% |
| Russia | 0.029 | 0.029 | 0.030 | 5.2% |
| USA | 0.006 | 0.007 | 0.005 | −23.3% |
| Vietnam | 0.059 | 0.048 | 0.069 | 45.0% |
| World | 0.019 | 0.018 | 0.020 | 16.4% |

* Processed from the data of World Bank and WSA.

Economists mentioned that developed countries had experienced a shift from industrial societies to services, information, and knowledge [32]. Thus, they incline to use steel and its derivative steel products more selectively [33].

The author of [34] examines the development of steel use intensity from fifteen European Union countries. He shows that the European Union's steel intensity had continuously saturated at 0.016 kg/USD in almost the last three decades and entered a stabilization stage. It means that currently, the European Union has entered a decade of services, information, and knowledge.

### 4.2. Correlation between Steel Intensity and Real GDP Per Capita

One of the key indicators that reflects economic development is GDP. The GDP measures the output of all goods and services in the economy [35]. The hypothesis states that a country's economic development level explains its intensity of metal use, including steel.

To know the relationship between the I-U of steel and real GDP per capita, we undertook the correlation between the two variables. There are various correlations between I-U and GDP per capita based on different economic conditions, according to [9]. We selected ten countries, including India, Vietnam, Indonesia, China, Malaysia, Russia, Korea, Japan, Germany, and the USA, with the data in Table 2 obtained from [20,31,36]. We selected those countries because they may represent the countries within the classification in terms of economic discrepancies. India, Indonesia, China, Russia, Japan, South Korea, Germany, and the USA are G20 countries. Besides that, we designated Vietnam because of its significant development in the steel industry. Malaysia can also be considered a country in transition to being a developed country.

**Table 2.** Comparison of Countries Steel-Related Data.

|  | India | Vietnam | Indonesia | China | Malaysia | Russia | South Korea | Japan | Germany | USA |
|---|---|---|---|---|---|---|---|---|---|---|
| Population (000) [36] | 1,366,418 | 96,462 | 270,626 | 1,397,715 | 31,950 | 144,374 | 51,709 | 126,265 | 83,133 | 328,240 |
| GDP per Capita (USD) [31] | 2257 | 3409 | 4333 | 12,556 | 11,109 | 12,195 | 34,998 | 39,313 | 51,204 | 70,249 |
| Steel Production (000 ton) [20] | 111,351 | 17,469 | 7783 | 996,342 | 6820 | 71,897 | 71,412 | 99,284 | 39,627 | 87,761 |

The correlations in Table 3 show that they vary from positive not significant, positive significant, and negative significant. The relationships are distinguished into lower-middle-income, upper-middle-income, and high-income countries from country selection data. The classification is determined based on The World Bank, as in Table 4, which assigns the world's economies to four income groups; low, lower-middle, upper-middle, and high-income countries based on Gross National Income (GNI) per capita [37]. GNI per capita is the USD value of a country's final income divided by its population in a year. It should reflect the average before-tax income of a country's citizens. GNI is related to GDP, comprising all income from investment abroad.

**Table 3.** Correlation between Steel IU and GDP per capita in selected countries.

|  | India | Vietnam | Indonesia | China | Malaysia | Russia | South Korea | Japan | Germany | USA |
|---|---|---|---|---|---|---|---|---|---|---|
| Pearson Correlation | 0.334 | 0.869 ** | 0.640 ** | −0.019 | −0.833 ** | 0.048 | −0.887 ** | −0.614 ** | −0.572 ** | −0.702 ** |
| Significant level | 0.129 | 0.000 | 0.001 | 0.932 | 0.000 | 0.833 | 0.000 | 0.002 | 0.005 | 0.000 |

* Correlation is significant at the 0.05 level (1-tailed). ** Correlation is significant at the 0.01 level (1-tailed). For the correlation with no *, it is not significant.

**Table 4.** Classification of countries based on income in USD.

| Group | 1 July 2020 (New) | 1 July 2019 (Old) |
|---|---|---|
| Low Income | GNI per capita < 1036 | GNI per capita < 1026 |
| Lower-middle income | 1036 ≤ GNI per capita < 4045 | 1026 ≤ GNI per capita < 3995 |
| Upper-middle income | 4046 ≤ GNI per capita < 12,535 | 3996 ≤ GNI per capita < 12,375 |
| High income | GNI per capita > 12,535 | GNI per capita > 12,375 |

Based on [28,31].

The data in Table 3 show that India has a positive but not significant correlation. It is different from Vietnam and Indonesia, which have positive and significant correlations. China has a negative and not significant correlation, while Russia has positive but not

significant correlations. Malaysia has a significant negative correlation, similar to Korea, Japan, Germany, and the USA.

From the classification, as in Table 5, the lower-middle-income countries are India and Vietnam. The upper-middle-income countries are Indonesia, China, Malaysia, and Russia, while the last three countries are transitioning from upper-middle-income to high-income. High-income countries are Korea, Japan, Germany, and the USA.

**Table 5.** Classification of countries based on nominal GNI per Capita.

| Country | GNI Per Capita (USD) | Group |
| --- | --- | --- |
| India | 2257 | Lower-middle income |
| Vietnam | 3409 | Lower-middle income |
| Indonesia | 4333 | Upper-middle income |
| China | 12,556 | Upper-middle income |
| Russia | 12,195 | Upper-middle income |
| Malaysia | 11,109 | Upper-middle income |
| South Korea | 34,998 | High income |
| Japan | 39,313 | High income |
| Germany | 51,204 | High income |
| USA | 70,249 | High income |

Based on [31].

### 4.3. Synthesis of the Correlations

From Appendix A, we reveal the correlations between I-U and GDP per capita, as shown in Table 6. It establishes a cycle of the waveform between correlation and significance in conjunction with the stages of economic development, as presented in Figure 3.

**Table 6.** Country Income and Correlations between Steel IU and GDP per Capita.

| Country | Low Income | Lower-Middle Income | Upper-Middle Income | Transition from Upper-Middle Income to High Income | High Income |
| --- | --- | --- | --- | --- | --- |
| India | | Positive Not significant | | | |
| Vietnam | | Positive Significant | | | |
| Indonesia | | | Positive Significant | | |
| China | No country in the classification of low-income develops its steel industry. | | | Negative Not Significant | |
| Malaysia | | | | Negative Significant | |
| Russia | | | Positive Not Significant | | |
| South Korea | | | | | Negative Significant |
| Japan | | | | | Negative Significant |
| Germany | | | | | Negative Significant |
| USA | | | | | Negative Significant |

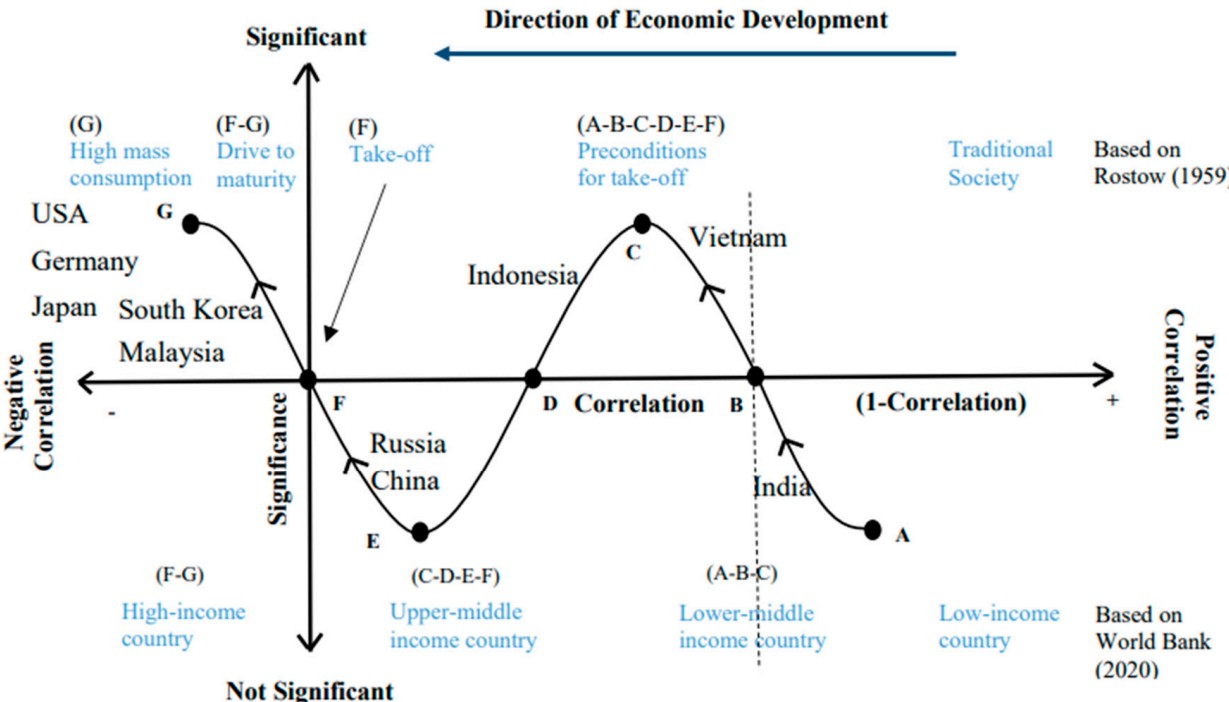

**Figure 3.** Cycle of Correlation and Significant between Steel I-U and real GDP per Capita along the country economic classification in 2022 based on [27,28].

Firstly, for the low-income country, relatively, there is no development of the steel industry. Therefore, we do not identify it in the cycle of the waveform. It can also refer to Korea's experience in the traditional society where industrial, such as steel, and commercial development was virtually non-existent [30]. Secondly, the lower-middle-income country has a positive correlation with both significant and not significant, a country whose economy has benefitted significantly from the steel industry's support, such as Vietnam [38]. In contrast, in India, the steel industry's existence has not yet significantly supported the economy; in other words, it has not revolutionized the industry in accordance with [27,39,40]. Therefore, we put the x-axis as (1–Correlation) for India as a fledgling lower-middle-income country. Thirdly, upper-middle-income countries have a positive correlation, both significant and not significant, except for countries in the stage of transitioning to high-income countries. Indonesia has a positive correlation because the growth of the economy is significantly supported by the steel industry [41]. China [42] has a slightly negative correlation, while Russia [43] has a positive correlation. The steel industry does not support the economies of the two countries substantially since they are initiating to switch industries to service and beginning to take advantage of technology to increase productivity. We place China on the left side of the x-axis despite the fact that there is a non-significant correlation because the GDP of China is the highest among all countries in the sample within the upper-middle-income. Malaysia has a negative correlation and significant [44]. We consider Malaysia as the country in the take-off stage (toward a high-income country) because the government has capitalized on the investment in technology to increase productivity and switch to service. Fourthly, a high-income country has a significant negative correlation. South Korea [45], Japan [46], Germany [47], and the USA [48] are the countries that concentrate on advancing technology to boost industry productivity and enhance the quality of products and services.

## 5. Linear Regression between I-U and Independent Variables

The results of significant components in the linear regression, as shown in Appendix A, are depicted in the following table.

From Table 7, we discover that there appears to be a pattern of significant components along the country's economic classification. For the lower-middle-income countries (India, Vietnam), the significant components are final consumption per capita. In terms of upper-middle-income countries (Indonesia, Russia, China, and Malaysia), the significant components are the import of goods and services per capita, investment per capita, and negative final consumption per capita. Related to high-income countries (South Korea, Japan, Germany, and the USA), the significant components are negative final consumption per capita and investment per capita. China and Malaysia can be regarded as countries in transition from upper-middle-income to high-income, where they also possess the component of higher income country (particularly negative final consumption per capita).

**Table 7.** Significant Components in the Linear Regression.

| India | Vietnam | Indonesia | Russia | China | Malaysia | South Korea | Japan | Germany | USA |
|-------|---------|-----------|--------|-------|----------|-------------|-------|---------|-----|
| Final Consumption per Capita | Final Consumption per Capita | Import of goods and services per Capita | Investment per Capita | Negative Final Consumption per Capita | Negative Final Consumption per Capita Investment per Capita | Negative Final Consumption per Capita | Negative Final Consumption per Capita | Negative Final Consumption per Capita | Negative Final Consumption per Capita Investment per Capita |

Related to the cycle of significances and correlations (in Figure 3), the finding of linear regression reinforces and matches Figure 3, where India and Vietnam (the lower-middle-income countries) are located on the right side of the cycle. In the middle side of the cycle, there are Indonesia, China, Russia, and Malaysia (upper-middle-income countries), with the exception that China and Malaysia are at the take-off level. On the left side of the cycle, there are high-income countries, such as South Korea, Japan, Germany, and the USA.

As a lower-middle-income country, India has a positive correlation between steel intensity and GDP per capita, but it is insignificant. In accordance with Rostow's model of development, India is still in the phase of preconditions for take-off. The agricultural contribution to GDP has reduced to less than twenty percent, and other sectors' contributions increased faster than agricultural production [40]. India's crop productivity is within these characteristics; the manufacturing industry has not developed with less use of technology. The top industries in India are textiles, chemicals, food processing, and steel [40].

Vietnam's condition is different from that of India, although they are both lower-middle-income countries. Vietnam has high steel intensity to support such infant industries as garment, food processing, shoes, and machine-building [38].

Indonesia is one of the upper-middle-income countries. The agriculture sector's contribution to GDP declined to 12.7% and was surpassed by industry, particularly manufacturing (19.7%), which required supporting steel consumption. The industry sector is the economy's most prominent and accounts for 46.4% of GDP, followed by services (38.6%) and agriculture (14.4%) [41].

Russia ranks 44 in the Global Finance Ranking of National Tech Strength [49]. Russia focuses on a complete range of mining and extractive industries producing coal, oil, gas, and chemicals [43]. Based on Rostow's model, Russia is in the preconditions for take-off.

China [42] and Malaysia [44], transitioning from upper-middle-income to high-income, invested heavily in technology [49]. At the same time, the I-U declined while steel production remained. The spillover effect that has occurred in the industry encouraged other steel-unrelated industries, such as services, to flourish [42,44]. China and Malaysia rank 32 and 33 in the Global Finance Ranking of National Tech Strength in 2022 [49]. The investment in technology spurred productivity that encouraged GDP growth. Besides that, it efficiently propelled the contribution of the industry. China focuses on mining and ore processing (iron, steel, aluminum, other metals, and coal), machine building, armaments, textiles and

apparel, petroleum, cement, and chemicals [42]. Malaysia has successfully diversified its economy from one that was initially agriculture and commodity-based to one that now plays host to robust manufacturing and service sectors, which have propelled the country to become a leading exporter of electrical appliances, parts, and components [44]. The Malaysian steel industry is largely regulated by the National Policy on Industry 4.0, which is concerned with the digital transformation of the manufacturing sector and its related services [50]. Based on Rostow's model, both countries are in the take-off stage.

The correlations are negative and significant for high-income countries, such as South Korea, Japan, Germany, and the USA. These suggest that the I-U in these countries decreased or has reached maturity [18,51] while they invested heavily in technology that spurs productivity in other industries. The decoupling and spillover effect have occurred in high-income countries. Ranking the Global Finance Ranking of National Tech Strength in 2022, these countries are Korea (1), Japan (7), Germany (13), and the USA (2) [41]. South Korea has leading advanced technology industries, including electronics, telecommunications, automobile production, and chemicals [45]. The situation is similar to Japan, with motor vehicles, electronic equipment, machine tools, steel, and nonferrous metals as leading industries [46].

In the cycle waveform, South Korea is at the point between F and G. Based on this position, South Korea is a high-income country in the stage of drive to maturity. We foresee that South Korea, in ten years, peaks the negative and significant correlation between IU and GDP per capita and enters into a stage of high mass consumption.

Between 1980 and 2000, the production of steel in Japan remained at about the same quantitative level, but in terms of GDP, it nearly doubled. At the same time, better steel products came into use in automobiles, buildings, and other fields [52].

Germany has both basic industries and cutting-edge technology industries, which complement national value [47]. The industries comprise iron, steel, coal, cement, chemicals, machinery, vehicles, machine tools, electronics, food and beverages, shipbuilding, and textiles [47]. Like Germany, the USA has petroleum, steel, motor vehicles, aerospace, telecommunications, chemicals, electronics, food processing, and consumer goods industries [48]. Besides that, they focus more on exporting products and services to export to the economy. The high-income countries switched industries from manufacturing to service, above 60% of the GDP.

Germany has adopted steel technologies relatively rapidly during the past years, especially in technologies that provide essential productivity benefits next to energy savings (e.g., BOF, CCM; see [53]).

In the cycle waveform, Germany is at the point of G. Based on the position, Germany is a high-income country in the stage of the age of high mass consumption. Germany will maintain its correlation between I-U and GDP per capita.

In the cycle waveform (Figure 3), the USA is at the point of G. Based on the position; the USA is a high-income country in the stage of the age of high mass consumption. We estimate that within ten years, the USA will maintain its correlation between I-U and GDP per capita.

From the discussions, we reveal the characteristics of country classification and stages of economic development as shown in Table 8.

From Table 6 and Figure 3, we plot the results to form the inverted U-shaped curve in the following Figure 4.

We discover that our revealed steel intensity is different from the common knowledge of previous research findings, as shown in Figure 1. The gradient between 1 and 2 is positive not significant (blue-dotted lines), between 2 and 3 is positive significant (red-dotted lines), while between 3 and 4 is positive not significant (blue-dotted lines), and between 4 to 5 is negative significant (green-dotted lines). Before take-off (in the previous study, it is the leveling off stage), there are three stages of transition from low-income to lower-middle-income (with no significant growth), lower-middle-income to upper-middle-income (with

significant growth), to take-off (with no significant growth or significant decline). From the take-off to high-income countries, the decline of I-U is immediately significant.

**Table 8.** Characteristics of Country Classification and Stage of Economic Development.

| Characteristics | Low Income Traditional Society | Lower-Middle Income Precondition for Take-Off | Upper-Middle Income Precondition for Take-Off | Transition from Upper-Middle Income to High Income Take-Off | High Income Drive to Maturity Age of High Mass Consumption |
|---|---|---|---|---|---|
| Economy | GNI per capita < 1036 | 1036 < GNI per capita < 4045 | 4046 < GNI per capita < 12,535 | Precondition for take-off < GNI per capita ≤ 12,535 | GNI per capita > 12,535 |
| Industry | -Dominated by agriculture with traditional cultivating forms -Productivity by man-hour work is lower compared to the subsequent growth stages | -Manufacturing industry has not been developed, and therefore the I-U is still developing -Agriculture sector dominance to GDP surpassed by manufacturing except for the country in the initial level of lower-middle-income -For specific countries, such as Vietnam, the government is so determined to develop manufacturing industries supported by the significant development of the steel industry | Spillover effect occurs in the industry: -Steel-related industries, such as automotive, real estate, machinery, and equipment, significantly supported GDP. -Heavy investment in technology while at the same time the steel intensity use decreases and steel production remained adequate for domestic production. | Investing in technology and orienting to export. | Focus on exporting the products and services contribution of export to the economy is prominent. High-income countries switched the industries from manufacturing to service |
| Steel Industry | -Steel and other related industries are not in existence yet | -Correlation between steel I-U and GDP per capita is positive and can be both significant and insignificant | Two conditions of the steel industry: -Correlation between steel I-U and GDP per capita is positive and significant -Correlation is positive insignificant The decoupling occurs between steel IU and GDP per capita | -Correlation between I-U and GDP per capita is negative significant -The decoupling occurs between steel I-U and GDP per capita -Concerned with the digital transformation of the manufacturing sector and its related services | -Correlation between steel I-U and GDP per capita is negative significant -The decoupling occurs between steel I-U and GDP per capita |

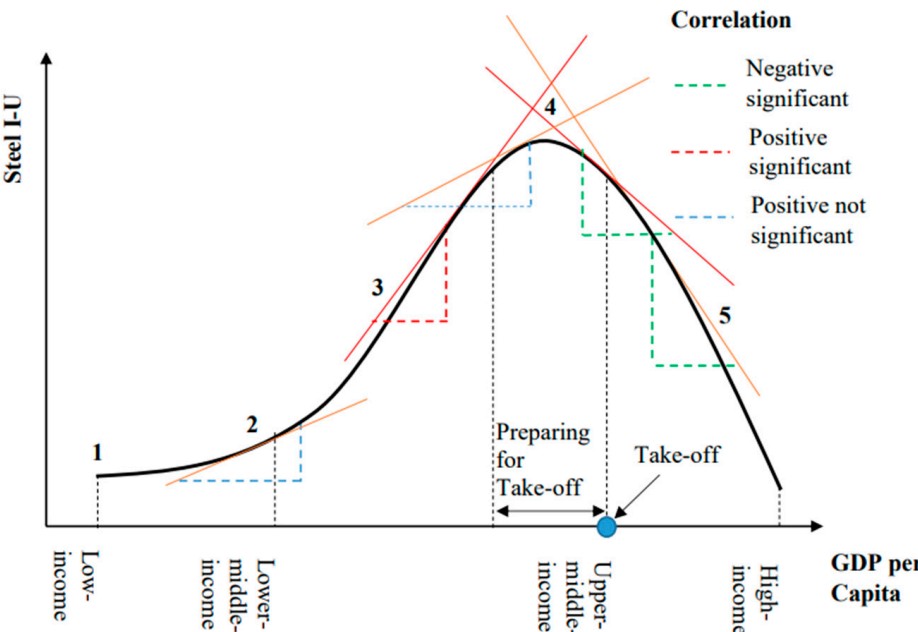

**Figure 4.** Plot of the Steel Intensity Curve.

### *5.1. Implications*

The implications derived from the correlations between I-U and GDP per capita are important for formulating government policy. We discern it into different economic development; low-income, lower-middle-income, upper-middle-income, a transition from upper-middle-income to high-income, and high-income.

### 5.1.1. Low-Income

Based on the results of correlation in accordance with the waveform and linear regression, in general, low-income countries should gradually enhance steel-related economic development. It will increase the correlation between GDP per Capita and I-U. Governments need to ameliorate the productivity of the agriculture industry [29]. A country within this classification shall develop basic infrastructure with steel-related supporting industries, such as roads, drainage systems, sewers, and electrical utilities [54]. The development of real-estate and large-scale infrastructure constructions may generate a large demand for low-end steel products, which becomes the preference since the investment return period is short, and the market benefits are undeniable. With government support, a large amount of capital will flow into the steel industry [55]. Therefore, during economic reform, the steel sector shall stand at the frontier of these reforms [23].

### 5.1.2. Lower-Middle-Income

Based on the previous results of correlation and linear regression, it can be implied that Iron and Steel is the basic industry and the leading industry to accelerate industrialization [56]. The government of lower-middle-income countries necessitates gradually enhancing the productivity of agriculture, forestry, and fishery industries [29] and simultaneously developing the manufacturing industry, particularly related with support to the agriculture industry. The steel industry shall enhance steel-supporting agriculture, construction, transportation, and electricity and gas supply industries. For instance, ports, roads, and railways airports must handle the massive cargo volumes entering and leaving the country. The development of power generation is necessary to prevent power outages for the industry. In this stage, anticipating and supporting industries' growth, reorganization, and steel industry consolidation is essential to possess high-quality steel enterprise resources. This possession is through more intense innovation and technology development with effective regulation enforcement [23]. The steel sector includes a series

of sub-industries: mine exploration, industrial design, project construction, technologies research, mining, coking, and supplementary materials. It contains complete subsystems and is managed like small communities, forming networks scattered across the country [23]. Before taking off, the countries must invest heavily in technology to encourage industry productivity [57].

### 5.1.3. Upper-Middle-Income

For upper-middle-income countries prepared for taking off to the drive to maturity stage, the government requisites to thrive in steel-related industries, such as automotive, real estate, machinery, and equipment. It aims to enhance and synergize steel-supporting agriculture, forestry, fishery, mining and quarrying, manufacturing, electricity, gas supply, construction, and transportation industries to bring national competitive advantage [58]. The technology aligns with the advancement of products or services and the process along with the value-chain activity. Countries require to exploit other markets and industries with sound supporting technology, industries, and productivity. Besides that, a global mindset by imposing on exports of products and services has a pivotal role in taking off.

Countries in the first stage of emergence as the upper-middle-income classification have a significant positive correlation between the use and GDP per capita. In this regard, the government shall develop the steel industry concomitant with the increase in I-U and advancement of the related industries' capacity.

As countries experience stagnant or declining I-U and increasing GDP per capita due to switching from manufacturing to service industries, they need to intensify technology to induce innovation. This declining steel intensity may thrive in other industries and transform and upgrade the economic structure. Therefore, it can release idle factors [55].

### 5.1.4. Transitioning Countries

The authors of [59,60] suggest that middle-income countries increase their R&D, human capital resources, international competitiveness, and dynamic comparative advantage and create high-quality institutions to avoid the middle-income trap. For countries transitioning to high-income, they need to invest heavily and capture cutting-edge technology [57]. Investments in education and innovation are essential for long-term growth to improve productivity, develop up-to-date products, and harness new markets abroad [60]. The capitalization of such investment may increase the proportion of export to GDP significantly. Furthermore, exploitation of service markets shall initiate to prevail along with the increasing value of the exported products.

### 5.1.5. High-Income

The government in high-income countries requires to bring the policy of enhancing leading-edge technology by relentlessly invigorating productivity, exploiting global markets, and increasing export. Given the comparative advantage, leading industries are a combination between basic industries and high-tech-based industries. The basic industries have cutting-edge supporting technology. Energy-saving, energy conservation, and waste elimination in production activity are steel industry characteristics with back-up from investment in technology [61]. Improvement in the quality of construction materials and industrial steel products supports other related industries. It is particularly more important for countries with a shortage of natural resources and their steel market relatively small. In such a way, for instance, iron and coke needed for steel melting also need to be imported from overseas [55]. Therefore, the country focuses on improving product quality and value-added steel products in this economic development stage rather than expanding steel production capacity.

*5.2. Lessons Learned from Upper-Middle-Income and High-Income Countries*

5.2.1. Preparing for Take-Off and Take-Off

The initial step for a country preparing for take-off is to apply modern scientific techniques to refurbish agriculture and industry functions. It aims at securing food supplies. The country has initiated manufacturing and steel-related industries since manufacturing has become more important, although not yet significant. In this step, I-U increases along with GDP, but not significantly.

In the later step for preparing for take-off, the country needs to significantly develop steel-related industries, such as automotive, real estate, machinery, and equipment. It aims to enhance and synergize steel-supporting agriculture, forestry, fishery, mining and quarrying, manufacturing, electricity, gas supply, construction, and transportation industries to bring national competitive advantage [58]. Agricultural productivity is still key, but the country concentrates only on thriving a few leading industries that support the economy's development to spur export from the back-up of heavy investment in technology. Since, in this step, the service embarks on providing its contribution to GDP. It may take 20 to 30 years to capitalize on the investment in technology [30]. The manufacturing and steel industries have become essential and present a significant contribution to GDP. The I-U significantly supports economic development. Gradually, those industries should absorb advanced and leading-edge technology to reduce energy consumption, improve productivity, and provide quality products and services. In general, a country needs to prepare for take-off to possess prior development of society and its economy, resulting in a positive, sustained, and self-reinforcing response [27].

For a country in the stage of taking off (transitioning), one or more substantial manufacturing sectors develop with a high rate of growth [30]. The overall steel industry is in the state of leveling off [18,51] and is imposed on increasing efficiency and productivity through intensive technology investment. The I-U declines or reaches maturity; therefore, the correlation with GDP per capita is significantly negative. It is also a result of the country switching more notably to service sectors. Industrialization is still the key, although the industry is shifting from light manufacturing to heavier industries. For instance, Korea diversified its industries and relied on ships, electronics, and iron and steel products [62]. An increase in exports should become a top priority.

Decoupling between I-U and GDP per capita is natural and even one of the benefits of the manufacturing sector, including steel, on the economy through the spillover effect and investment in innovation [13].

5.2.2. Drive to Maturity

In this stage, the country relies heavily on export and investment in technology [57]. It concentrated only on a few leading industries related to the core competencies leading to national competitive advantage [58]. New leading sectors are gathering momentum to supplant the older leading sectors of the take-off, where deceleration has increasingly slowed the pace of expansion [27]. Although the service sector continues to grow, manufacturing remains prevalent, often challenging manufacturing as the leading sector [30].

The I-U decreases or reaches maturity [18,51] while investing heavily in innovation, research, and development spurred productivity in industries. The USA and South Korea's experiences suggest that investment in research and development increased to 3.47 percent of GDP.

5.2.3. Age of High Mass Consumption

In this stage, the country concentrates more on exporting products and services, which confirms the prominent role of export in the economy. The high-income countries switch industries from manufacturing to service to above 60% of the GDP [63].

The industries comprise basic and cutting-edge technology industries [64], which complement national value.

The I-U declined because of investment in technology [18,51], which spurred productivity and focused massively on service sectors. In South Korea, agriculture makes up a mere 3 percent of the GDP and employs 7.2 percent of the labor. Industry constitutes 39.4 percent of GDP, employing 25.1 percent of the labor force, while the service sector makes up 57.6 percent of GDP and employs more than two-thirds of the labor force [30]. In 1960, agriculture accounted for 40% of the Korean GDP [65].

## 6. Conclusions

Based on this paper, a plan for economic development supported by knowledgeable concerns about the relationship between steel I-U and GDP per Capita with its components as depicted in the cycle of the waveform may provide an elucidated trajectory of developing industries and transforming the economy.

The authors of [6–11] mention that there is a correlation between steel I-U and the GDP per Capita; however, they neither specifically elaborate on the components of GDP per Capita along with the economic development nor the country's economic stratification, let alone the contribution of steel, for example, to the required or aspired industry development.

The concept of economic growth and the identification within society of modernization's sequential economic steps is acknowledged by [27]. However, specifically, there is a lack of attention and investigation towards the steel industry's role and further linkages, spillovers, and externalities to the industries, which may differ among different countries with specific economic stratification.

This paper has elaborated on the I-U and its implication for economic development. We unraveled the significance of the correlations between the steel I-U and economic development in the cycle of a waveform, which is discerned into different economic stratification based on [27,28]. For this purpose, we have extended the previous research of [6–11]. Our research also reinforces the previous aforementioned literature by unraveling the relationship of components between GDP and the steel I-U and elucidates them in the cycle of the waveform. Subsequently, we also related economic development and stratification with the steel industry and its related industry development. In this revelation, our research streamlines [27] the economic development model, which may present a comprehensive trajectory for a country's economic development.

Our revealed steel intensity curve (Figure 4) is different from the common knowledge of previous research findings, as shown in Figure 1. Before take-off (in the previous study, it is the leveling off), there are three stages of transition from low-income to lower-middle-income (with a not significant correlation), lower-middle-income to upper-middle-income (with positive significant correlation), to take-off (with positive not significant correlation or negative significant correlation). From the take-off to a high-income country, the decline of I-U is immediately toward a negative significant correlation between I-U and GDP per Capita.

In regard to the purpose of this research, firstly, related to how significant the correlations between steel I-U and economic development, the paper reveals discrepancies among nations with different economic conditions regarding the correlation between I-U and GDP per Capita.

We concur with the study result of [9], who was the recent scholar investigating the steel I-U, and that in middle-income panels, both the intensity of steel use variable and the GDP per capita variable are non-stationary. However, we extend and rectify the previous research of [9] that the non-stationary state is not only in the middle-income panel but the transition of all phases of economic development from the low-income, lower-middle-income, upper-middle-income, to high-income-countries, are non-stationary. We reveal that there are distinctive significant correlations among different economic development; even the middle-income countries transitioning from low-income to lower-middle-income require revolutionization (non-stationary), as depicted in the waveform as we revert the x-axis from (1-correlation) to correlation. As such, middle-income countries transitioning

from lower-middle-income to higher-middle-income countries up to take-off are in different places of waveform, which means they are non-stationary.

Relatively, there is no development of the steel industry in a low-income country. Therefore, we do not identify the position in the cycle of the waveform. Referring to Figure 3 about the waveform, the lower-middle-income country has a positive correlation with both significant and not significant. Upper-middle-income countries have a positive correlation, both significant and not significant, except for a country in the stage of transitioning to a high-income country with a significant negative correlation. A high-income country has a negative correlation and is significant.

Secondly, aligned with the results of the correlation and the linear regression in terms of enhancing the economy and related steel industries, the country with different economic classifications has its specific industries to be focused on.

For low-income countries, the steel-related industries aim to develop infrastructure of roads, drainage systems, sewers, and electrical utilities to bolster the agriculture industry.

Lower-middle-income countries shall develop and enhance the productivity of the agriculture, forestry, and fishery industries [29] and simultaneously build the manufacturing industry [66,67], particularly related with support to the agriculture industry. Industrialization is the most important engine of economic growth [30]. The steel industry shall enhance steel-supporting agriculture, construction, transportation, and electricity and gas supply industries.

Upper-middle-income countries need to develop and synergize such steel-related industries as automotive, real estate, machinery, equipment, agriculture, forestry and fishery, mining and quarrying, manufacturing, electricity and gas supply, construction, and transportation industries [66,67]. It aims to achieve a national competitive advantage [58] supported by competencies and capabilities [68] by shifting and increasing various-related industries [69]. Before taking off, the countries shall invest fervently in technology since R&D expenditures represent the main engine of technological progress [69,70] and economic growth [57,71–73]. Increasing various-related industries enhance the proportion of exports to GDP [74].

High-income countries shall possess leading industries, which are a combination between basic industries and high-tech-based industries. The basic industries have cutting-edge supporting technology. Energy-saving, energy conservation, and waste elimination in production activity are steel industry characteristics with the back-up from investment in technology [61]. Improvement in the quality of construction materials and industrial steel products supports other related industries.

Our unveiled pattern of the waveform (Figure 3) suggests that each country classification has a different economic policy to enhance the economy related to the required steel or non-steel-related industry. The classification is determined by the World Bank, and we connect it with Rostow's economic development. We also reveal that correlation and linear regression results matched and reinforced.

Limitations or challenges may occur in terms of sampling of countries using judgmental sampling that can implicate biases or shortcomings in the data collection process or the reliability of the data sources. Besides that, the determination of countries prior to the revolutionized phase from point A-B in the waveform, such as India in the sample, shall be meticulously examined.

**Author Contributions:** Resources, Murtyastanto; Writing—original draft, H.W. All authors have read and agreed to the published version of the manuscript.

**Funding:** This research received no external funding.

**Data Availability Statement:** The economic data were obtained from the World Bank. The Steel Data were from the World Steel Association. Both are in the list of references. The Steel-Intensity-of-Use data were derived from both economic and steel data.

**Conflicts of Interest:** The authors declare no conflict of interest.

# Appendix A

## India (Endogenous)

### Coefficients[a]

| Model | | Unstandardized Coefficients B | Std. Error | Standardized Coefficients Beta | t | Sig. |
|---|---|---|---|---|---|---|
| 1 | (Constant) | 0.035 | 0.002 | | 18.522 | 0.000 |
| | GDP per Capita | $2.198 \times 10^{-6}$ | 0.000 | 0.334 | 1.583 | 0.129 |

a. Dependent Variable: Steel Intensity

### Model Summary

| Model | R | R Square | Adjusted R Square | Std. Error of the Estimate |
|---|---|---|---|---|
| 1 | 0.334[a] | 0.111 | 0.067 | 0.00263973 |

a. Predictors: (Constant), GDP per Capita

## India (Endogenous with Exogenous)

### Coefficients[a]

| Model | | Unstandardized Coefficients B | Std. Error | Standardized Coefficients Beta | t | Sig. |
|---|---|---|---|---|---|---|
| 1 | (Constant) | 0.034 | 0.002 | | 18.774 | 0.000 |
| | Final Consumption per Capita | $-6.912 \times 10^{-6}$ | 0.000 | -1.283 | -2.910 | 0.010 |
| | Investment per Capita | $1.217 \times 10^{-5}$ | 0.000 | 0.955 | 1.338 | 0.198 |
| | Export of Goods and Services per Capita | $1.551 \times 10^{-5}$ | 0.000 | 0.957 | 0.737 | 0.471 |
| | Import of Goods and Services per Capita | $-1.878 \times 10^{-6}$ | 0.000 | -0.135 | -0.105 | 0.918 |

a. Dependent Variable: Steel Intensity

### Model Summary

| Model | R | R Square | Adjusted R Square | Std. Error of the Estimate |
|---|---|---|---|---|
| 1 | 0.835[a] | 0.698 | 0.627 | 0.00166945 |

a. Predictors: (Constant), Import_of_Goods_and_Services_per_Capita, Final_Consumption_per_Capita, Investment_per_Capita, Export_of_Goods_and_Services_per_Capita

## Vietnam (Endogenous)

### Coefficients[a]

| Model | | Unstandardized Coefficients B | Std. Error | Standardized Coefficients Beta | t | Sig. |
|---|---|---|---|---|---|---|
| 1 | (Constant) | 0.025 | 0.009 | | 2.877 | 0.012 |
| | GDP per Capita | $1.636 \times 10^{-5}$ | 0.000 | 0.774 | 4.736 | 0.000 |

a. Dependent Variable: Steel Intensity

### Model Summary

| Model | R | R Square | Adjusted R Square | Std. Error of the Estimate |
|---|---|---|---|---|
| 1 | 0.774[a] | 0.599 | 0.573 | 0.00834378 |

a. Predictors: (Constant), GDP per Capita

## Vietnam (Endogenous with Exogenous)

### Coefficients[a]

| Model | | Unstandardized Coefficients B | Std. Error | Standardized Coefficients Beta | t | Sig. |
|---|---|---|---|---|---|---|
| 1 | (Constant) | -0.026 | 0.017 | | -1.540 | 0.150 |
| | Final Consumption per Capita | $8.389 \times 10^{-5}$ | 0.000 | 2.384 | 2.984 | 0.011 |
| | Investment per Capita | $-1.450 \times 10^{-6}$ | 0.000 | -0.020 | -0.026 | 0.979 |
| | Export of Goods and Services per Capita | $-4.547 \times 10^{-5}$ | 0.000 | -2.484 | -2.067 | 0.061 |
| | Import of Goods and Services per Capita | $1.917 \times 10^{-5}$ | 0.000 | 0.921 | 0.858 | 0.408 |

a. Dependent Variable: Steel Intensity

### Model Summary

| Model | R | R Square | Adjusted R Square | Std. Error of the Estimate |
|---|---|---|---|---|
| 1 | 0.898[a] | 0.807 | 0.742 | 0.00647908 |

a. Predictors: (Constant), Import_of_Goods_and_Services_per_Capita, Final Consumption per Capita, Investment per Capita, Export of Goods and Services per Capita

## Indonesia (Endogenous)

### Coefficients[a]

| Model | | Unstandardized Coefficients B | Std. Error | Standardized Coefficients Beta | t | Sig. |
|---|---|---|---|---|---|---|
| 1 | (Constant) | 0.010 | 0.001 | | 7.880 | 0.000 |
| | GDP per Capita | $1.553 \times 10^{-6}$ | 0.000 | 0.640 | 3.722 | 0.001 |

a. Dependent Variable: Steel Intensity

### Model Summary

| Model | R | R Square | Adjusted R Square | Std. Error of the Estimate |
|---|---|---|---|---|
| 1 | 0.640[a] | 0.409 | 0.380 | 0.001349720 |

a. Predictors: (Constant), GDP per Capita

## Indonesia (Endogenous with Exogenous)

### Coefficients[a]

| Model | | Unstandardized Coefficients B | Std. Error | Standardized Coefficients Beta | t | Sig. |
|---|---|---|---|---|---|---|
| 1 | (Constant) | 0.006 | 0.003 | | 1.982 | 0.064 |
| | Final Consumption per Capita | $-2.080 \times 10^{-6}$ | 0.000 | -0.514 | -1.134 | 0.272 |
| | Investment per Capita | $3.463 \times 10^{-6}$ | 0.000 | 0.655 | 1.234 | 0.234 |
| | Export of Goods and Services per Capita | $9.796 \times 10^{-7}$ | 0.000 | 0.032 | 0.162 | 0.874 |
| | Import of Goods and Services per Capita | $1.277 \times 10^{-5}$ | 0.000 | 0.703 | 2.456 | 0.025 |

a. Dependent Variable: Steel Intensity

### Model Summary

| Model | R | R Square | Adjusted R Square | Std. Error of the Estimate |
|---|---|---|---|---|
| 1 | 0.883[a] | 0.780 | 0.729 | 0.000892852 |

a. Predictors: (Constant), Import of Goods and Services per Capita, Final Consumption per Capita, Investment per Capita, Export of Goods and Services per Capita

## China (Endogenous)

### Coefficients[a]

| Model | | Unstandardized Coefficients B | Std. Error | Standardized Coefficients Beta | t | Sig. |
|---|---|---|---|---|---|---|
| 1 | (Constant) | 0.066 | 0.005 | | 13.921 | 0.000 |
| | GDP per Capita | $-6.150 \times 10^{-8}$ | 0.000 | -0.019 | -0.087 | 0.932 |

a. Dependent Variable: Steel Intensity

### Model Summary

| Model | R | R Square | Adjusted R Square | Std. Error of the Estimate |
|---|---|---|---|---|
| 1 | 0.019[a] | 0.000 | -0.050 | 0.00938832 |

a. Predictors: (Constant), GDP per Capita

## China (Endogenous with Exogenous)

### Coefficients[a]

| Model | | Unstandardized Coefficients B | Std. Error | Standardized Coefficients Beta | t | Sig. |
|---|---|---|---|---|---|---|
| 1 | (Constant) | 0.058 | 0.006 | | 10.043 | 0.000 |
| | Final Consumption per Capita | $-1.653 \times 10^{-5}$ | 0.000 | -2.834 | -3.039 | 0.007 |
| | Investment per Capita | $1.191 \times 10^{-5}$ | 0.000 | 1.709 | 1.250 | 0.228 |
| | Export of Goods and Services per Capita | $1.330 \times 10^{-5}$ | 0.000 | 0.725 | 0.917 | 0.372 |
| | Import of Goods and Services per Capita | $9.607 \times 10^{-6}$ | 0.000 | 0.477 | 0.416 | 0.683 |

a. Dependent Variable: Steel Intensity

### Model Summary

| Model | R | R Square | Adjusted R Square | Std. Error of the Estimate |
|---|---|---|---|---|
| 1 | 0.844[a] | 0.712 | 0.645 | 0.00546299 |

a. Predictors: (Constant), Import of Goods and Services per Capita, Final Consumption per Capita, Investment per Capita, Export of Goods and Services per Capita

**Figure A1.** *Cont.*

### Russia (Endogenous)

**Coefficients[a]**

| | | Unstandardized Coefficients | | Standardized Coefficients | | |
|---|---|---|---|---|---|---|
| Model | | B | Std. Error | Beta | t | Sig. |
| 1 | (Constant) | 0.029 | 0.003 | | 8.856 | 0.000 |
| | GDP per Capita | $8.073 \times 10^{-8}$ | 0.000 | 0.048 | 0.213 | 0.833 |

a. Dependent Variable: Steel Intensity

**Model Summary**

| Model | R | R Square | Adjusted R Square | Std. Error of the Estimate |
|---|---|---|---|---|
| 1 | 0.048[a] | 0.002 | -0.048 | 0.00263531 |

a. Predictors: (Constant), GDP per Capita

### Russia (Endogenous with Exogenous)

**Coefficients[a]**

| | | Unstandardized Coefficients | | Standardized Coefficients | | |
|---|---|---|---|---|---|---|
| Model | | B | Std. Error | Beta | t | Sig. |
| 1 | (Constant) | 0.034 | 0.006 | | 5.488 | 0.000 |
| | Final Consumption per Capita | $-3.379 \times 10^{-6}$ | 0.000 | -1.531 | -1.975 | 0.065 |
| | Investment per Capita | $1.133 \times 10^{-5}$ | 0.000 | 1.919 | 2.703 | 0.015 |
| | Export of Goods and Services per Capita | $-4.285 \times 10^{-7}$ | 0.000 | -0.044 | -0.121 | 0.905 |
| | Import of Goods and Services per Capita | $-2.929 \times 10^{-6}$ | 0.000 | -0.283 | -0.232 | 0.819 |

a. Dependent Variable: Steel Intensity

**Model Summary**

| Model | R | R Square | Adjusted R Square | Std. Error of the Estimate |
|---|---|---|---|---|
| 1 | 0.639[a] | 0.408 | 0.269 | 0.00220164 |

a. Predictors: (Constant), Import of Goods and Services per Capita, Final Consumption per Capita, Investment per Capita, Export of Goods and Services per Capita

### Malaysia (Endogenous)

**Coefficients[a]**

| | | Unstandardized Coefficients | | Standardized Coefficients | | |
|---|---|---|---|---|---|---|
| Model | | B | Std. Error | Beta | t | Sig. |
| 1 | (Constant) | 0.066 | 0.005 | | 13.827 | 0.000 |
| | GDP per Capita | $-3.614 \times 10^{-6}$ | 0.000 | -0.833 | -6.722 | 0.000 |

a. Dependent Variable: Steel Intensity

**Model Summary**

| Model | R | R Square | Adjusted R Square | Std. Error of the Estimate |
|---|---|---|---|---|
| 1 | 0.833[a] | 0.693 | 0.678 | 0.00411401 |

a. Predictors: (Constant), GDP per Capita

### Malaysia (Endogenous with Exogenous)

**Coefficients[a]**

| | | Unstandardized Coefficients | | Standardized Coefficients | | |
|---|---|---|---|---|---|---|
| Model | | B | Std. Error | Beta | t | Sig. |
| 1 | (Constant) | 0.072 | 0.012 | | 6.085 | 0.000 |
| | Final Consumption per Capita | $-6.908 \times 10^{-6}$ | 0.000 | -1.493 | -7.676 | 0.000 |
| | Investment per Capita | $8.963 \times 10^{-6}$ | 0.000 | 0.532 | 2.207 | 0.041 |
| | Export of Goods and Services per Capita | $-5.495 \times 10^{-6}$ | 0.000 | -0.402 | -1.464 | 0.161 |
| | Import of Goods and Services per Capita | $3.651 \times 10^{-6}$ | 0.000 | 0.197 | 0.720 | 0.481 |

a. Dependent Variable: Steel Intensity

**Model Summary**

| Model | R | R Square | Adjusted R Square | Std. Error of the Estimate |
|---|---|---|---|---|
| 1 | 0.921[a] | 0.848 | 0.812 | 0.00314277 |

a. Predictors: (Constant), Import of Goods and Services per Capita, Final Consumption per Capita, Investment per Capita, Export of Goods and Services per Capita

### South Korea (Endogenous)

**Coefficients[a]**

| | | Unstandardized Coefficients | | Standardized Coefficients | | |
|---|---|---|---|---|---|---|
| Model | | B | Std. Error | Beta | t | Sig. |
| 1 | (Constant) | 0.069 | 0.003 | | 21.125 | 0.000 |
| | GDP per Capita | $-1.091 \times 10^{-6}$ | 0.000 | -0.887 | -8.607 | 0.000 |

a. Dependent Variable: Steel Intensity

**Model Summary**

| Model | R | R Square | Adjusted R Square | Std. Error of the Estimate |
|---|---|---|---|---|
| 1 | 0.887[a] | 0.787 | 0.777 | 0.00281477 |

a. Predictors: (Constant), GDP per Capita

### South Korea (Endogenous with Exogenous)

**Coefficients[a]**

| | | Unstandardized Coefficients | | Standardized Coefficients | | |
|---|---|---|---|---|---|---|
| Model | | B | Std. Error | Beta | t | Sig. |
| 1 | (Constant) | 0.070 | 0.004 | | 16.998 | 0.000 |
| | Final Consumption per Capita | $-2.968 \times 10^{-6}$ | 0.000 | -1.484 | -2.629 | 0.018 |
| | Investment per Capita | $1.902 \times 10^{-6}$ | 0.000 | 0.472 | 1.022 | 0.321 |
| | Export of Goods and Services per Capita | $-1.571 \times 10^{-6}$ | 0.000 | -0.811 | -1.038 | 0.314 |
| | Import of Goods and Services per Capita | $2.219 \times 10^{-6}$ | 0.000 | 1.031 | 1.553 | 0.139 |

a. Dependent Variable: Steel Intensity

**Model Summary**

| Model | R | R Square | Adjusted R Square | Std. Error of the Estimate |
|---|---|---|---|---|
| 1 | 0.924[a] | 0.854 | 0.820 | 0.00252644 |

a. Predictors: (Constant), Import of Goods and Services per Capita, Final Consumption per Capita, Investment per Capita, Export of Goods and Services per Capita

### Japan (Endogenous)

**Coefficients[a]**

| | | Unstandardized Coefficients | | Standardized Coefficients | | |
|---|---|---|---|---|---|---|
| Model | | B | Std. Error | Beta | t | Sig. |
| 1 | (Constant) | 0.046 | 0.009 | | 5.305 | 0.000 |
| | GDP per Capita | $-8.946 \times 10^{-7}$ | 0.000 | -0.614 | -3.483 | 0.002 |

a. Dependent Variable: Steel Intensity

**Model Summary**

| Model | R | R Square | Adjusted R Square | Std. Error of the Estimate |
|---|---|---|---|---|
| 1 | 0.614[a] | 0.378 | 0.346 | 0.00181804 |

a. Predictors: (Constant), GDP per Capita

### Japan (Endogenous with Exogenous)

**Coefficients[a]**

| | | Unstandardized Coefficients | | Standardized Coefficients | | |
|---|---|---|---|---|---|---|
| Model | | B | Std. Error | Beta | t | Sig. |
| 1 | (Constant) | 0.074 | 0.015 | | 4.832 | 0.000 |
| | Final Consumption per Capita | $-2.776 \times 10^{-6}$ | 0.000 | -1.889 | -4.334 | 0.000 |
| | Investment per Capita | $5.925 \times 10^{-8}$ | 0.000 | 0.014 | 0.091 | 0.928 |
| | Export of Goods and Services per Capita | $4.542 \times 10^{-7}$ | 0.000 | 0.221 | 0.519 | 0.611 |
| | Import of Goods and Services per Capita | $1.618 \times 10^{-6}$ | 0.000 | 0.955 | 1.593 | 0.130 |

a. Dependent Variable: Steel Intensity

**Model Summary**

| Model | R | R Square | Adjusted R Square | Std. Error of the Estimate |
|---|---|---|---|---|
| 1 | 0.872[a] | 0.760 | 0.703 | 0.00122474 |

a. Predictors: (Constant), Import of Goods and Services per Capita, Final Consumption per Capita, Investment per Capita, Export of Goods and Services per Capita

**Figure A1.** *Cont.*

**Germany (Endogenous)**

**Coefficients[a]**

| Model | | Unstandardized Coefficients | | Standardized Coefficients | t | Sig. |
|---|---|---|---|---|---|---|
| | | B | Std. Error | Beta | | |
| 1 | (Constant) | 0.022 | 0.003 | | 6.928 | 0.000 |
| | GDP per Capita | $-2.502 \times 10^{-7}$ | 0.000 | -0.572 | -3.122 | 0.005 |

a. Dependent Variable: Steel Intensity

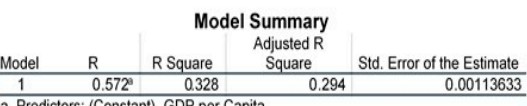

**Germany (Endogenous with Exogenous)**

**Coefficients[a]**

| Model | | Unstandardized Coefficients | | Standardized Coefficients | t | Sig. |
|---|---|---|---|---|---|---|
| | | B | Std. Error | Beta | | |
| 1 | (Constant) | 0.048 | 0.009 | | 5.554 | 0.000 |
| | Final Consumption per Capita | $-1.711 \times 10^{-6}$ | 0.000 | -2.299 | -3.934 | 0.001 |
| | Investment per Capita | $1.077 \times 10^{-7}$ | 0.000 | 0.071 | 0.215 | 0.832 |
| | Export of Goods and Services per Capita | $1.767 \times 10^{-7}$ | 0.000 | 0.436 | 0.295 | 0.772 |
| | Import of Goods and Services per Capita | $6.158 \times 10^{-7}$ | 0.000 | 1.238 | 0.716 | 0.484 |

a. Dependent Variable: Steel Intensity

**Model Summary**

| Model | R | R Square | Adjusted R Square | Std. Error of the Estimate |
|---|---|---|---|---|
| 1 | 0.572[a] | 0.328 | 0.294 | 0.00113633 |

a. Predictors: (Constant), GDP per Capita

**Model Summary**

| Model | R | R Square | Adjusted R Square | Std. Error of the Estimate |
|---|---|---|---|---|
| 1 | 0.778[a] | 0.605 | 0.512 | 0.00094496 |

a. Predictors: (Constant), Import of Goods and Services per Capita, Final Consumption per Capita, Investment per Capita, Export of Goods and Services per Capita

**USA (Endogenous)**

**Coefficients[a]**

| Model | | Unstandardized Coefficients | | Standardized Coefficients | t | Sig. |
|---|---|---|---|---|---|---|
| | | B | Std. Error | Beta | | |
| 1 | (Constant) | 0.022 | 0.004 | | 4.921 | 0.000 |
| | GDP per Capita | $-3.113 \times 10^{-7}$ | 0.000 | -0.639 | -3.423 | 0.003 |

a. Dependent Variable: Steel Intensity

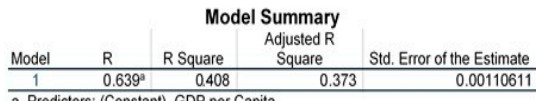

**USA (Endogenous with Exogenous)**

**Coefficients[a]**

| Model | | Unstandardized Coefficients | | Standardized Coefficients | t | Sig. |
|---|---|---|---|---|---|---|
| | | B | Std. Error | Beta | | |
| 1 | (Constant) | 0.022 | 0.002 | | 10.432 | 0.000 |
| | Final Consumption per Capita | $-6.241 \times 10^{-7}$ | 0.000 | -1.181 | -9.171 | 0.000 |
| | Investment per Capita | $8.050 \times 10^{-7}$ | 0.000 | 0.541 | 7.262 | 0.000 |
| | Export of Goods and Services per Capita | $2.496 \times 10^{-9}$ | 0.000 | 0.002 | 0.012 | 0.991 |
| | Import of Goods and Services per Capita | $4.102 \times 10^{-7}$ | 0.000 | 0.295 | 1.960 | 0.070 |

a. Dependent Variable: Steel Intensity

**Model Summary**

| Model | R | R Square | Adjusted R Square | Std. Error of the Estimate |
|---|---|---|---|---|
| 1 | 0.639[a] | 0.408 | 0.373 | 0.00110611 |

a. Predictors: (Constant), GDP per Capita

**Model Summary**

| Model | R | R Square | Adjusted R Square | Std. Error of the Estimate |
|---|---|---|---|---|
| 1 | 0.972[a] | 0.946 | 0.930 | 0.00036967 |

a. Predictors: (Constant), Import of Goods and Services per Capita, Final Consumption per Capita, Investment per Capita, Export of Goods and Services per Capita

**Figure A1.** Linear Regression of Endogenous and Exogenous Parameters.

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
