# Peer review of "The Implication of Steel-Intensity-of-Use on Economic Development"

_sustainability, doi:10.3390/su151612297_

Round 1

Reviewer 1 Report

The paper addresses a fascinating and appealing research topic; the approach and the topics discussed in the paper are new and justify the publication's interest. The paper's content is good, and the presentation of information is quite clear. Some considerations could be evaluated before final acceptance:

Abstract:

- The methodology of the research work is not stated in the abstract.

Introduction:

- The manuscript's overall contribution needs to be better explored: what exactly is the key finding of this paper? Which are the research questions addressed by the study? What exactly is new? To this end, the discussion needs to build explicit links to particular literature that is challenged, extended, or otherwise influenced by this study.

- The literature review should be updated to enhance, broaden, and deepen state of the art. The study is focused on the relationships between the Steel Intensity of Use and the Gross Domestic Product per Capita; are the problems highlighted well-known in the literature? How have these issues been faced? An investigation of the existing strategies is missing. The author doesn’t consider the available scientific literature on the approaches already adopted to evaluate these issues.

- The aim of the paper stated in the introduction is not consistent with the purpose stated in the summary of the work.

- A brief presentation of the framework of the manuscript is required.

Conclusion

- A synthesis of major gaps filled by the manuscript and further development of the approach used could help to focus results of the scientific investigation carried out and propose future development;

Framework:

- Please re-arrange the structure of the paper. For instance,  the section “Theory” (probably the name should be changed) should be included before “Materials and Methods” and not vice-versa. Please avoid introducing new sub-sections to describe the five growth stages. In my opinion, these are uncommon and favour unnecessary text fragmentation. Moreover, it is not clear the reason for which these subsections were identified with two names.

Figures, tables, and equations:

- Figures 2 and 3 are incomplete and fragmented. Therefore, they are difficult to read.

- The caption of the tables shown in the appendix of the manuscript is missing.

Appendix:

- No reference was provided in the body of the manuscript to the tables shown in the appendix.

Minor editing of the English language required.

Author Response

Dear Valued Reviewer,

Thank you very much for your comments

  1. Manuscript’s overall contribution

- What exactly is the key finding of this paper?

This research reveals the relationship between the I-U and the GDP, differentiated into economic development stratification which may differ from the previous studies. It also describes the economy, general industry, and steel industry and unravels I-U's required roles to develop the industries and transform the economy within economic development, which are not undertaken by the previous studies.

- Which are the research questions addressed by the study?

The research questions addressed by this study:

How significantly does I-U in the related steel industries influence economic development?

Are there any discrepancies among nations with different economic conditions regarding I-U's influences on economic development?

What are industries, in relationships with the I-U and the GDP per Capita, to be developed in a country with particular economic development?

- What exactly is new?

This research reveals the relationship between the I-U and the GDP, differentiated into economic development stratification which may differ from the previous studies. It also describes the economy, general industry, and steel industry and unravels I-U's required roles to develop the industries and transform the economy within economic development which are not undertaken by the previous studies.

The discussion has presented explicit links to particular literature that is challenged, extended, or otherwise influenced by this study (In this concern, we place it in the conclusion), as the following:

The I-U hypothesis has been tested on several metals, for example steel [9], aluminium, copper, lead, zinc and nickel [14,15,16,12,17,18,19,20]. Despite the existence of literatures on I-U, however, the particular research on relationship between I-U with the GDP per capita in terms of stratification of economic development (from low-income-country to high-income country) are still non-existent. The gaps are particularly, in terms of steel industry roles on economic development and influences to other industries, industries differences among countries within similar phase of development, and required steel industry and related industries development projection based on correlations and significance.

- Are the problems highlighted well-known in the literature?

Yes, the problems highlighted are well known in literatures in terms of the I-U hypothesis about the relationship between I-U and the GDP per Capita. These issues have been faced by various scholars on metal I-U including aluminium, copper, lead, zinc and nickel.

We mention that a notable study on steel I-U was conducted by Wårell (2014), however, the particular research on relationship between I-U with the GDP per capita in terms of stratification of economic development (from low-income-country to high-income country) are still non-existent. The gaps are particularly, in terms of steel industry roles on economic development and influences to other industries, industries differences among countries within similar phase of development, and required steel industry and related industries development projection based on correlations and significance.

- The aim of the paper stated in the introduction is not consistent with the purpose stated in the summary of the work

We incorporate the aim of the paper and the purpose with the identified gaps based on the literature on I-U. In this concern, we inserted “which may differ from the previous studies” and “which are not undertaken by the previous studies”, to make it consistent among the literature, aim, and purpose.

- A brief presentation of the framework of the manuscript is required.

We use the following sentence to develop the framework in the forthcoming section of Model framework for analysis of steel consumption intensity.

“Steel is one of the prominent metals as it is widely used to develop industries so there is a correlation between steel consumption and economy [10]. Steel Intensity-of-Use follows the hypothesis. It has an inverted U-shaped along with GDP. There is a point of GDP per capita as the peak where the intensity of use starts to decline [11]. This condition is due to the fact that I-U for a country is a function of its product composition of output which in turn depends on the GDP per Capita [12], while components GDP comprise personal consumption, investment, government spending, and net exports [13]”.

  1. Conclusion

We have included a synthesis of major gaps filled by the manuscript as mentioned in the introduction and further development of the approach used to incorporate the aim, purpose, and novelty of the findings.

  1. Framework

- The section “Theory” (probably the name should be changed) should be included before “Materials and Methods” and not vice-versa.

We have exchanged the place of theory section with “Materials and Methods. We have also change the term “theory” into “literature review”.

- Please avoid introducing new sub-sections to describe the five growth stages. In my opinion, these are uncommon and favour unnecessary text fragmentation.

We have integrated the sub-sections into the respective sections.

- It is not clear the reason for which these subsections were identified with two names.

We have deleted the first name related to the World Bank Economic Classification and focused on the terms from Rostow (1959).

  1. Figures, tables, and equations:

- Figures 2 and 3 are incomplete and fragmented. Therefore, they are difficult to read.

We have completed several issues in the figures (Please see in the current Figure 2 and Figure 3).

In order to provide clarity to the readers related with the I-U, we have also made the new Figure 4.

- The caption of the tables shown in the appendix of the manuscript is missing.

We have inserted the caption of “As in Appendix A”.

  1. Appendix:

- No reference was provided in the body of the manuscript to the tables shown in the appendix.

We have inserted the sentence: “All data of the I-U and GDP per Capita including the components were processed as in Appendix A”.

Sincerely,

Harimukti Wandebori

Reviewer 2 Report

My comments on the submitted manuscript:

1.                  While the abstract covers essential points, it could benefit from providing more specific details about the methodology used and the key findings obtained.

2.                  It would be beneficial to provide a brief explanation of how this research contributes to the existing literature and why it is important to unravel the roles of I-U in developing industries and transforming the economy.

3.                  The introduction briefly mentions that the Steel Intensity of Use follows an inverted U-shaped relationship with GDP per capita. Elaborate on the factors or mechanisms that drive this relationship and clarify the point at which the intensity of use starts to decline, providing more clarity to the reader.

4.                  Consider briefly mentioning the specific years covered in the analysis to provide clarity on the temporal scope of the study.

5.                  Consider briefly discussing any potential limitations or challenges encountered during the data collection or analysis process to acknowledge any potential biases or constraints in the study.

6.                  The formatting of the manuscript needs to be improved.

Minor corrections are required.

Author Response

Dear Valued Reviewer,

Thank you very much for your comments.

  1. We have inserted the methodology and key findings in the abstract, as the following:

This paper identifies statistical relationships between the Steel Intensity of Use and the Gross Domestic Product per Capita based on the data of ten selected countries based on judgment sampling from the World Steel Association. Economic data such as Gross Domestic Product, Gross National Income, Gross Domestic Product per capita, Government Spending, Investment, Export of Goods and Services, and Manufacturing are obtained. Based on regression analysis and exploration of economic data, the relationship differs in terms of economic development stratification. We find a pattern of a waveform in terms of Correlation and Significance between Steel Intensity of Use and the Gross Domestic Product per Capita along the Country Economic Classification. Each classification implies for the country to enhance the economy related to the required steel or non-steel related industry and the accompanying government policy.

  1. Brief explanation on how this research contributes to the existing literatures can be obtained in the introduction after discussing and reviewing them, here in the following sentences.

Despite the existence of literatures on I-U, however, the particular research on relationship between steel I-U with the GDP per capita in terms of stratification of economic development (from low-income-country to high-income country) are still non-existent. The gaps are particularly, in terms of steel industry roles on economic development and influences to other industries, industries differences among countries within similar phase of development, and required steel industry and related industries development projection based on correlations and significance.

This research reveals the relationship between the I-U and the GDP, differentiated into economic development stratification which may differ from the previous studies. It also describes the economy, general industry, and steel industry and unravels I-U's required roles to develop the industries and transform the economy within economic development.

  1. In the introduction, we elaborate on the factors or mechanisms that drive this relationship and clarify the point at which the intensity of use starts to decline, in order to provide more clarity to the reader.

Steel is one of the prominent metals as it is widely used to develop industries so there is a correlation between steel consumption and economy [10]. Steel Intensity-of-Use follows the hypothesis. It has an inverted U-shaped along with GDP. There is a point of GDP per capita as the peak where the intensity of use starts to decline [11]. This condition is due to the fact that I-U for a country is a function of its product composition of output which in turn depends on the GDP per Capita [12], while components GDP comprise personal consumption, investment, government spending, and net exports [13].

  1. We briefly mentioning the specific years covered in the analysis to provide clarity on the temporal scope of the study.

In section Materials and Methods, we acknowledge the specific years of data of published steel consumption and production data from the World Steel Association (WSA) from 2000 to 2021 [32].

In Table 2: Comparison of Countries Steel-Related Data, we mention year of 2022 in the references

In Table 4, in terms of Country Economic Classification we use the most recent World Bank Classification in 2020 and compared with 2019.

In Figure 3 about the waveform, we use World Bank Classification in 2020 based on economic data of 2022.

       5.  We briefly discuss any potential limitations or challenges encountered during the data collection or analysis process to acknowledge any potential biases or constraints in the study in the last sentence of the article as the following:

Limitations or challenges may occur in terms of sample of countries using judgment sampling that can implicate biases or shortcomings in the data collection process or the reliability of the data sources. Besides that, in the determination of country prior to revolutionized phase from point A-B in the waveform, such as India in the sample, shall be meticulously examined.

  1. We have improved several formatting issues, such as referencing and improving figures.

Sincerely,

Harimukti Wandebori

Reviewer 3 Report

Dear Author,

please read and refer to the comments in the text. I read the text carefully and found it valuable.

Please also consider the structure of the article. In the current version, it seems to me that you have divided the content in too much detail. The article contains as many as 26 separate points (1. Introduction; 2. Materials and methods; 3. Theory; 3.1. Steel consumption and economics; 3.2. I-U; 3.3. Model framework for analysis of steel consumption intensity; 3.4.1 Low income/traditional society 3.4 .2.Medium-Low Income/Start Condition;3.4.3.Middle Income/High Growth;3.4.4.High Income/Striving for Maturity or Age of High Consumption;4 Scores 4.1 I-U in developing and developed countries 4.2 Correlation between Intensity steel and real GDP per capita 4.3 Correlation synthesis 5 Linear regression between I-U and independent variables 5.1 Implications 5.1.1 Low income 5.1.2 Low average income 5.1 3 Higher average income 5.1.4 Transition countries 5.1.5 High income 5.2. Lessons learned from middle-income countries; 5.2.1. preparation for launch and launch; 5.2.2. striving for maturity; 5.2.3. Age of high mass consumption; 6. Conclusion) - this is far too much. Please link the threads to make it easier to understand the research perspective.

I would like to see the changes - I'd love to read the text again.

Kind regards,

Author Response

Dear Valued Reviewer,

Thank you very much for your comments.

We have already abridged the number of content division.

We have confined the sub-division to only one, i.e. 2, 2.1, 2.2, 2.3, and 2.4. The rest of the sub-divisions are embedded into the main division.

To make it easier to understand, we exchange section 2 (Materials and methods) with section 3 (Theory). We also change the term “Theory” in previous section 2 into “Literature Review”.

Our abridged version is as the following.

(1. Introduction; 2. Literature Review; 2.1. Steel consumption and economics; 2.2.I-U; 2.3. Model framework for analysis of steel consumption intensity; 2.4 Country Classification and Economic Stages of Development 3. Materials and methods, 4. Results 4.1. I-U in developing and developed countries 4.2 Correlation between Intensity steel and real GDP per capita 4.3 Correlation synthesis 5 Linear regression between I-U and independent variables 5.1 Implications; 5.2. Lessons learned from middle-income countries; 6. Conclusion)

Currently, the article contains as many as 15 separate points.

We have linked the threads. In the introduction; we inserted some literatures to emphasize that there is a gap in the current literatures regarding relationship between I-U with the GDP per capita in terms of stratification of economic development (from low-income-country to high-income country), which are still non-existent. Particularly, in terms of steel industry roles on economic development and influences to other industries, industries differences among countries within similar phase of development, and required steel industry and related industries development projection based on correlations and significance. We also emphasize the main purpose and the research revelation link with the aforementioned literature gap. The conclusion in this article refers to the main purposes and the contribution to fill the gap identified in the introduction.

Sincerely,

Harimukti Wandebori

Round 2

Reviewer 1 Report

The paper addresses a very interesting and appealing research topic; the approach and the topics discussed in the article are new and justify the interest in the publication. The structure of the paper is correct. The revisions adopted have improved the work. The suggestions proposed in the review report are included in the last draft of the paper. The major gaps filled by the manuscript are now described in “Literature review”, “Results”, and “Conclusions”. The mistakes identified in the previous draft of the paper have been corrected.

Good Luck!

Author Response

Dear Reviewer,

Thank you very much for all the comments and suggestions.

We are very pleased that all have been fulfilled.

Regards,

Harimukti Wandebori

Reviewer 3 Report

Dear Author,

please refer to comment 2: "please check the results for China (Table 3 shows a negative correction for China). The entire Results section needs to be corrected.. Please refer to this comment - without this it is difficult to review the rest of the article."

I still think that the text is well written and interesting - it has publication potential. However, mistakes in the results section must be corrected before publication in MDPI-journal.

Kind regards,

Author Response

Dear Reviewer,

Thank you very much for your comments.

Regarding the first comment related to the structure of this manuscript with the main purpose to elaborate the I-U and its implication on economic development, we add a paragraph:

“The revealed correlation and its significant level is distinguished based on the respective country’s economic development. Its pattern is depicted as a waveform in terms of correlation and significance between Steel Intensity of Use and the Gross Domestic Product per Capita along the Country Economic Classification. The position of a country in the waveform implies the roles of steel industry to the economic development, how it transforms to the related industries, and switches to the service industry along with the economic development”.

Regarding the second comment related with China’s correlation and significant level:

We have fixed the result that China has a negative and not significant correlation, therefore, we apply it to the entire content of this manuscript.

Table 6 has been fixed accordingly; China has negative and not significant level of correlation. We also move China to the country of transitioning from upper-middle income to high-income.

We also have revised this paragraph below Table 6.

“China [46] has slightly negative correlation, while Russia [47] has positive correlation. The steel industry does not support the economies the two countries substantially, since they are initiating to switch the industries to service and beginning to take advantage of technology to increase productivity. We place China on the left side of the x-axis despite the fact that there is a not-significant correlation, because the GDP of China is the highest among all countries in the sample within the upper-middle-income”.

We have moved China, Figure 3 (revised version), adjacent to Malaysia.

We notify the revised items with red font and use a separate file since I have difficulty inserting Figure 3 to the ‘sustainability file’.

Regards,

Harimukti Wandebori

Round 3

Reviewer 3 Report

Dear Authors,

I am satisfied with the corrections you have made. As I wrote, I am not a fan of such a methodology, please consider my comments in your future texts.

Regards,